# Signaling Roleplay between Ion Channels during Mammalian Sperm Capacitation

**DOI:** 10.3390/biomedicines11092519

**Published:** 2023-09-12

**Authors:** Filip Benko, Dana Urminská, Michal Ďuračka, Eva Tvrdá

**Affiliations:** 1Institute of Biotechnology, Faculty of Biotechnology and Food Sciences, Slovak University of Agriculture in Nitra, Tr. A. Hlinku 2, 949 76 Nitra, Slovakia; dana.urminska@uniag.sk (D.U.); eva.tvrda@uniag.sk (E.T.); 2AgroBioTech Research Centre, Slovak University of Agriculture in Nitra, Tr. A. Hlinku 2, 949 76 Nitra, Slovakia; michal.duracka@uniag.sk

**Keywords:** ion channels, spermatozoa, capacitation, hyperpolarization, acrosome reaction

## Abstract

In order to accomplish their primary goal, mammalian spermatozoa must undergo a series of physiological, biochemical, and functional changes crucial for the acquisition of fertilization ability. Spermatozoa are highly polarized cells, which must swiftly respond to ionic changes on their passage through the female reproductive tract, and which are necessary for male gametes to acquire their functional competence. This review summarizes the current knowledge about specific ion channels and transporters located in the mammalian sperm plasma membrane, which are intricately involved in the initiation of changes within the ionic milieu of the sperm cell, leading to variations in the sperm membrane potential, membrane depolarization and hyperpolarization, changes in sperm motility and capacitation to further lead to the acrosome reaction and sperm–egg fusion. We also discuss the functionality of selected ion channels in male reproductive health and/or disease since these may become promising targets for clinical management of infertility in the future.

## 1. Introduction

Ion channels play a significant role in the regulation of membrane potential by maintaining intracellular pH (pH_i_), osmotic balance as well as sperm physiological responses associated with fertilization such as hyperactivated motility, capacitation, chemotaxis, and acrosome reaction. An increase in pH_i_ and alkalization of sperm cytosol is essential for hyperpolarization of the plasmatic membrane and later hyperactivation of sperm motility via a Ca^2+^-dependent pathway. Membrane potential (Em) or resting membrane potential (Em_r_) reflects different concentrations of ions (mmol/L) between intra- and extra-cellular spaces [1,2,3]. During capacitation, sperm cells change membrane potential by the process called hyperpolarization, which increases a negative charge of membrane through to reduced permeability for Na^+^ and increased permeability for K^+^. In non-capacitated mammalian spermatozoa, the value of Em varies from −35 to −45 mV while the Em of capacitated cells is around −65 mV [4,5].

Sperm ion channels are pore-forming proteins which can be found in the whole surface of the cell including principal piece or midpiece of flagellum and head (Figure 1). They are classified based on their opening or closing into voltage-gated and ligand-gated ion channels [6]. The regulation of voltage-gated ion channels depends on the voltage gradient across the plasmatic membrane, charge (cation/anion) or species of ions (Na^+^, Ca^2+^, H^+^, Cl^−^, K^+^). On the other hand, the activity of ligand-gated channels is managed through to the specific bind of primary signaling transmitters such as cyclic nucleotides [7,8]. The ability of spermatozoa to undergo capacitation depends on numerous factors like membrane potential, pH homeostasis and balanced ion environment. Mutual cooperation between ion channels, pumps and transporters is required for proper sperm motility. From all ions, Ca^2+^ is one of the most crucial because its higher concentrations (100–300 nM) start the hyperactivation of spermatozoa. In mammals, Ca^2+^ participate in the activation of the signaling pathways of capacitation as a secondary messenger. Before capacitation itself, sperm cells received specific signals from environment in the female genital tract. It all starts with the increase in intracellular pH and uptake of bicarbonate, which stimulate the sAC/cAMP/PKA pathway as we mention below. From the point of view of ion channel activity, intracellular alkalinization managed the activity of CatSper and KSper channels, which is required for capacitation [9,10]. In this paper, we provide an overview of the most important ion channels occurring in spermatozoa and analyze their involvement in sperm activation following their entry into the female reproductive system, including the process of capacitation, hyperactivation and acrosome reaction. The review methodology is available in Appendix A.

## 2. Bicarbonate Transporters

Bicarbonate transporters or acid extruders (Table 1) can maintain the intracellular pH homeostasis by transporting HCO_3_^−^, which induce phosphorylation of functional flagellar proteins in serine, threonine, and tyrosine residues through to the sAC/cAMP/PKA pathway. Soluble adenylate cyclase (sAC) catalyzes the synthesis of cAMP, which is sensitive for higher concentration of HCO_3_^−^. In general, HCO_3_^−^ ions are responsible for (1) the initiation of sperm motility right after ejaculation and (2) the activation of sperm capacitation in the female reproductive tract. The range of pH in the seminal fluid is between 7.2 and 8.4; it is believed that seminal fluid works as a buffer which controls the acidic environment of the vagina. Bicarbonate membrane transporters are represented by two major protein groups, solute carrier 4 (SLC4) and solute carrier 26 (SLC26) [11,12,13].

The principal group of the SLC4 family can be divided based on affinity to Na^+^-independent and Na^+^-dependent HCO_3_^−^ exchangers (Na^+^/HCO_3_^−^ cotransporters—NBCs). Na^+^-independent transporters maintain electroneutral exchange of Cl^−^ into HCO_3_^−^, which is ensured by three anion exchangers (AE) SLC4A1 (AE1), SLC4A2 (AE2), SLC4A3 (AE3) and two Na^+^-coupled exchangers SLCA48 (NDCBE), SLCA49 (AE4). The subfamily of Na^+^-dependent HCO_3_^−^ exchangers includes two electrogenic SLC4A4 (NBCe1), SLC4A5 (NBCe2) and two electroneutral SLC4A7 (NBCn1), SLC4A10 (NCBE) exchangers. The functionality of electroneutral transporters depends on the chemical gradient between intra- and extra-cellular space compared to electrogenic transporters, which depends on the negative electrical potential of the membrane [14,15,16]. The main core of the SLC4 channel is made from 14 loop-connected transmembrane α-helices (TM1–14) and amphipathic helices (H1–6) which form the core domain (TM1–4/TM8–11) and the gate domain (TM5–7/TM12–14). The active site of SLC4 is localized on the N-terminus of half-helices TM3 and TM10 while the blocker of the active side takes place at segment TM8 [17]. Several sources confirm that NBC channels, especially electrogenic ones, play an essential part in sperm capacitation, hyperactivated motility and membrane hyperpolarization through the initial fast increase in bicarbonate, which is necessary for cAMP/PKA pathway activation and later redistribution of cholesterol [3,18].

A group of SLC26 contains 11 electrogenic anion channels, but only 5 of them are able to transport HCO_3_^−^, represented by SLC26A3, SLC26A4, SLC26A6, SLC26A7 and SLC26A9. Like their cousins, the SLC26 channels are made from 14 transmembrane α-helices connected by loops, which occasionally contain amphipathic helices (H). The active site is also in the N-termini of TM3 and TM10 half-helices, but the C-termini region contains STAS (sulphate transporter anti-sigma factor antagonist domain) or the dimerization domain, involved in the expression and protein interaction, which can interact with the regulatory R-domain of the CFTR (cystic fibrosis transmembrane conductance regulator) channel. TM1–4 together with TM8–11 helices form the core domain, while TM5–7 and TM12–14 represent the gate [19,20]. The functional interaction between SLC26 and CFTR channels in capacitated spermatozoa regulates and maintains the high bicarbonate entrance, the sAC/PKA pathway and acrosomal exocytosis together with the ions of Ca^2+^. On the other hand, the inhibition of SLC26, especially SLC26A3 and SLC26A6 localized in the midpiece of the flagellum, can cause a decrease in the Cl^−^ influx and a blockage of intracellular alkalization and membrane hyperpolarization [21,22].

**Table 1 biomedicines-11-02519-t001:** Bicarbonate transporters.

Channel	Species	Localization	Functionality	References
SLCA26A3/A6	human, mice, guinea pig	sperm plasma membrane, acrosomal region of sperm head	HCO_3_^−^ transport, pH_i_ alkalinization, protein phosphorylation, hyperactivation, CFTR channel interactions	[19,23,24]
SLCA26A8	human, mice	equatorial segment of sperm	regulation of sperm motility and acrosome exocytosis during capacitation, CFTR channel interactions	[25]
SLC4A1	human, mice	sperm head and flagellum	actin depolymerization and regulation of acrosome reaction	[26]

## 3. Sodium Channels

Overall, spermatozoa are exposed to high sodium concentrations in the female reproductive tract, which directly regulate sperm membrane potential and electrogenic Na^+^/K^+^ ATPase gradient pump. An increased Na^+^ influx has a great influence on membrane polarization and improves linear sperm motility. The presence of voltage-gated Na^+^ channels (VGNCs) was confirmed in the human and bovine sperm cells (Table 2). Their activation depends on the depolarization of the sperm plasma membrane and the conduction of sodium ions into the cell, which directly supports the action potential [27,28].

Based on the structure, the core of the VGNC channel includes a big α subunit made from four repeat homologous domains (RD1–RD4) with six transmembrane helices (TM1–TM6) as well as one or more auxiliary β subunits (β1–β4), each consisting of a big extracellular N-terminal domain and an intracellular tail depending on the isoform. Furthermore, TM4, known as a positively charged sensor, manages the channel ion permeability by moving into the extracellular space and Na^+^ is transported via the pore P-loop between TM5 and TM6. Repeat domains play an important role in the PKA/PKC protein phosphorylation due to the connection with long intracytoplasmic loops. The level of protein phosphorylation of RD loops regulates the inactivation of the channel. The members of the VGNC family were localized particularly in the sperm flagellum (Na_V_1.2, Na_V_1.6, Na_V_1.8 and Na_X_) and the connecting piece (Na_V_1.4, Na_V_1.7 and Na_V_1.9) due to the immunostaining of human spermatozoa. In the case of Na_V_1.8, its presence was confirmed in bull and ram sperm. These channels have the ability to maintain progressive motility rather than hyperactivation or acrosome reaction [29,30,31].

The electrogenic transport of Na^+^ as well as membrane potential in spermatozoa can be also regulated by heteromultimeric epithelial sodium selective channels (ENaCs). They come from a superfamily of non-voltage-gated and amiloride-sensitive DEG/ENaC ion channels [32]. The channel itself is formed from four subunits (α, β, γ and δ), where α and δ are characterized as pore forming subunits which can be found in the midpiece or the principal piece of the sperm flagellum (ENaC-α) and the acrosome region (ENaC-δ). The activity of ENaC is controlled by intracellular pH, Ca^2+^, Cl^−^, phosphorylation or amiloride. Previous findings support the involvement of this channel in sperm movement and regulation of sperm resting potential, which becomes more negative during capacitation-associated hyperpolarization [33,34,35].

**Table 2 biomedicines-11-02519-t002:** Sodium channels.

Channel	Species	Localization	Functionality	References
VGNC, Na_v_	human, bull	sperm connecting piece and flagellum	maintaining of progressive motility, tyrosine phosphorylation	[27,28]
ENaC	human	central region of sperm flagellum, acrosome	regulation of sperm motility and sperm resting potential during capacitation, activator of CFTR channel, membrane hyperpolarization	[33]

## 4. Calcium Channels

As secondary intracellular messengers of capacitation, ions of calcium (Ca^2+^) participate in acrosome reaction, tyrosine phosphorylation, modulation of the cAMP-dependent pathway and maintenance of functionality of mitochondria as well as synthesis of ATP, which is necessary for hyperactivated motility [7]. The concentration of Ca^2+^ can be regulated by alcium channels (Table 3), transporters, or exchangers like CatSper (cation channel of sperm), VGCCs (voltage-gated Ca^2+^ channels), TRPVs (transient receptor potential vanilloids), SOCCs (store-operated Ca^2+^ channels) and CNGs (cyclic nucleotide-gated channels) [36]. Intracellular Ca^2+^ levels are also regulated by Ca^2+^ pumps of the plasma and outer acrosomal membranes, as well as mitochondrial transporters.

In general, the heterotetrameric complex of the CatSper channel is composed from four main pore-forming alpha subunits (CatSper 1–4) and six minor ancillary subunits CatSper β (beta), γ (gamma), δ (delta), ε (epsilon), ζ (zeta) and EFCAB9 (calcium-binding domain-containing protein 9). The activity of these channels is related to hyperactivation, sperm chemotaxis and thermotaxis as well as acrosome reaction. Every α subunit is made from six transmembrane domains (TM 1–6), which form a voltage-sensing domain (TM1–4) containing voltage sensors and a pore-forming region (TM 5–6), which particularly coordinate the Ca^2+^ influx. The CatSper channel can be found in the principal piece of the sperm flagellum in the form of four-sided longitudinal nanodomains responsible for sperm motility [37]. However, the principal piece of flagellum is missing organelles, so there is the theory that the CatSper complex is part of the membrane in the principal piece, and it is involved in the regulation of flagellar movement [38]. The pH sensible small voltage-gated CatSper channel is primary activated by the intracellular alkalization and the pH is regulated via a special histidine-rich region of the N-terminus part of CatSper1, but its activity can be also controlled by cyclic nucleotides, phosphorylation, progesterone, prostaglandins, glycoproteins of zona pellucida or bovine serum albumin as part of the oviductal fluid [39,40]. According to Hwang et al. [41], the cytoplasm complex CatSper ζ together with the EFCAB9 subunit work as gatekeepers and activity modulators of the other CatSper domains. The genes, which coded the functions of CatSper channels, are expressed in the testis during spermatogenesis.

Another major functional subclass of Ca^2+^ permeable channels are voltage-gated Ca^2+^ channels (VGCC’s) or voltage-sensitive Ca^2+^ channels (Ca_V_), which can be divided into high- and low-voltage-activated (HVA/LVA) channels localized in the sperm tail. Their activation depends on the changes in membrane potential and strong/low depolarization. Regulatory mechanisms of Ca_V_ channels could be activated by protein kinases (protein kinases A/C, Ca^2+^/calmodulin (CaM)-dependent protein kinase II) or ions of Ca^2+^ itself. Similar to CatSper channels, VGCCs contain four main transmembrane domains (1–4), but every central pore is surrounded by six transmembrane α helixes (S1–6). The α_1_ subunit of the central pore is encoded at least 10 genes from three subfamilies Ca_V_1–3. Based on their different physiological properties, several types of VGCCs were identified in the spermatozoa, including the L-type (Ca_V_1.1–1.4), T-type (Ca_V_3.1–3.3), R-type (Ca_V_2.3), and P/Q-type (Ca_V_2.1), which are involved in motility hyperactivation, capacitation, acrosome reaction as well as phosphorylation of the cAMP-dependent protein kinase A [42,43].

Transient receptor potential vanilloid channels (TRPVs) belong to a big branch of cation polymodal voltage-gated and Ca^2+^ permeable cellular channels (TRPV1–6), which are activated by the increased intracellular Ca^2+^ concentration via acrosomal inositol triphosphate (IP3) receptors or phospholipase C through receptor-mediated messenger phosphatidylinositol 4,5-bisphosphate (PIP_2_). These channels are responsible for the regulation of sperm functionality at different levels like basal and hyperactivated motility, thermotaxis, or acrosome reaction [44,45]. Interestingly, TRPV channels work as modulators of the Ca^2+^ signaling pathway of other Ca^2+^-permeable channels, which include regulations of the cytosolic cation flux and electrical activity because of the unique gate mechanisms and a wide range of ion selectiveness. Based on their structure, TRPV channels consist of six transmembrane segments (S1–6) and a pore-creating intracellular loop between the S5 and 6 segments [46,47].

Non-voltage-dependent store-operated Ca^2+^ channels (SOCCs) can operate under a negative membrane potential when VGCCs stay inactive. Their basal structure is formed by ORAI proteins (ORAI1–3) found in the sperm head and flagellum where each protein contains four TMs which create a pore (between TM2 and TM3)- and Ca^2+^-binding domain (CBD) localized in the center of the pore. This domain basically regulates the activity of the channel by binding Ca^2+^ or CaM, which inactivate the channel. The main role of the SOCC channel is to bind and store the extracellular Ca^2+^ in sperm mitochondria as well as regulate the motility and the acrosome reaction [48,49,50]. SOCC is activated when the intraacrosomal Ca^2+^ levels are reduced due to Ca^2+^ efflux from the acrosome via the inositol trisphosphate receptor (IP3-R). The activity of SOCCs can be inhibited with the induction of AMPK (5′ AMP-activated protein kinase) phosphorylation, which leads into reduction in the asymmetrical flagellar beating necessary for chemotaxis [51].

Cyclic nucleotide-gated cation channels (CNGs) are ligand voltage-gated channels, which use the free binding energy of second messengers like cAMP (cyclic adenosine monophosphate) or cGMP (cyclic guanosine monophosphate) for the regulation of pore opening [8,52]. Their structure is made from a heterotetrameric complex including homologous A (CNGA1–4) subunits, which defines principal channel properties and B (CNGB1/3) subunits necessary for gating kinetics. Similar to other voltage-gated ion channels, A as well as B subunits contain six transmembrane α helices (S1–6) and an ion-selective pore loop between S5 and S6. A cyclic nucleotide-binding domain (CNDB) formed at the cytosolic C-terminus of a channel is responsible for its activation [53]. In mammalian spermatozoa, CNG channels are important modulators of motility, capacitation, and acrosome reaction because of high permeability to Ca^2+^ and cGMP. The activity of these channels can be inhibited by Mg^2+^ or Ca^2+^ itself by binding to calmodulin, which also acts as a voltage-dependent blocker of permeability for monovalent cations including Na^+^ and K^+^ [54].

**Table 3 biomedicines-11-02519-t003:** Calcium channels.

Channel	Species	Localization	Functionality	References
CatSper	mice, human, boar, bull, sea urchin	principal piece	promotion of hyperactivated motility, sperm chemotaxis and thermotaxis, late acrosome reaction, Ca^2+^ uptake, alkalinization	[55,56,57]
VGCC, Ca_v_	newt, marine fish, equine, mice, bull, human	sperm neck and tail	regulation of total and progressive motility, hyperactivation, capacitation, phosphorylation of protein kinase A, acrosome reaction	[58,59,60,61]
TRPV	vertebrates	whole surface of spermatozoa	activation of basic and hyperactivated motility, capacitation, membrane depolarization, opening of other channels (CatSper, Hv1)	[45,62,63]
SOOC	mice, chicken, ascidian	sperm head and flagellum	regulation of sperm motility and acrosome reaction, induction of 5′ AMP-activated protein kinase (AMPK) phosphorylation	[50,51,64]
CNG	mammals, sea urchin	flagellum of spermatozoa	effectors for CNG induced Ca^2+^ response, sperm hyperactivation	[65]

## 5. Proton Channels

A specific group of channels involved in carrier-mediated proton transport are combined membrane sodium–hydrogen exchangers/antiporters (NHE’s) and the Hv1 voltage-gated ion channel. NHEs are membrane proteins encoded by the solute carrier 9 (*SLC9*) gene family which transport Na^+^ into the cell and H^+^ out of the cell across the lipid bilayer accompanied with the maintenance of intracellular pH_i_ and Na^+^ homeostasis (Table 4). Several members of NHEs were identified, especially in spermatozoa. NHE1 and NHE10/sNHE (sperm-specific NHE isoform with a binding site for cAMP) expressed in the principal piece of the sperm flagellum are important for normal sperm motility and capacitation, while NHE8 ensures the formation of the acrosome. Knock-out of any of these NHEs results in male infertility characterized by a lower expression of sAC and intracellular cAMP, which confirms the mutual relationship with the cAMP signal pathway [66,67]. NHEs are formed from 12 TM helices connected together with six extracellular (EL1–6) and five intracellular loops (IL1–5). The channel also contains cytosolic N- and C-terminal domini with an extracellular N-linked glycosylation site [68].

The Hv1 pH-sensitive channel is co-localized together with other flagellar channels like CatSper and KSper in the principal piece, and it is activated by membrane depolarization, alkaline changes in the extracellular environment or the removal of zinc, which works as a potential blocker of Hv1 activity. Based on their similar subcellular location, proton exchange through Hv1 promotes intraflagellar alkalinization and stimulates CatSper channels in human or bovine spermatozoa compared to murine spermatozoa, which lacks the presence of Hv1; its role in pH regulation is replaced by an Na^+^-dependent Cl^−^/HCO_3_^−^ exchanger or sNHE. The synergy pathway between the Hv1/CatSper channels elevates intracellular pH (5.5–6.5) as well as the Ca^2+^ uptake, which is necessary for the activation of pH-dependent axonemal proteins and the maintenance of sperm motility, chemotaxis, capacitation, and later acrosome reaction [61,69,70]. The architecture of an Hv1 dimeric channel complex resembles voltage-gated channels, with the presence of voltage sensor domain (VSD) but missing a separate pore domain. Instead of that, the Hv1 channel contains internal selective proton transfer water wire located in the center of VSD [71].

**Table 4 biomedicines-11-02519-t004:** Proton channels.

Channel	Species	Localization	Functionality	References
NHE1	human, ram, mice, rat, boar	mid- and principal piece of sperm flagellum	activation of sperm motility upon ejaculation, co-activation of Ca^2+^ and SLO channels	[72,73,74]
NHE10/sNHE	human, mice	principal piece of flagellum	regulation of cAMP signal pathway, hyperactivated motility, sAC regulation	[66,75]
Hv1	human, bull, boar	principal and terminal part of sperm tail close to CatSper	promotion of fast intracellular alkalinization via pH_i_ regulation, synergic pathway with CatSper channels for the regulation of capacitation and late acrosome reaction	[3,61,76]

## 6. Potassium Channels

The primary role of potassium (K^+^) channels (Table 5) in male gametes is the hyperpolarization of the plasma membrane during the capacitation process, which is a crucial step for the sperm motility hyperactivation [7]. The opening of K^+^ channels leads to changes in the membrane potential in the capacitated sperm [4], resulting in membrane hyperpolarization, whilst the closure of these channels causes its depolarization. The first indication of their existence was reported by Arnoult et al. [77], who observed that membrane hyperpolarization that accompanies sperm capacitation is affected by the external K^+^ concentration and K^+^-channel blockers, assuming that the hyperpolarization process can be executed by the opening of these channels.

Four major classes of K^+^ channels are currently recognized, specifically (1) voltage-gated K^+^ channels which open or close depending on fluctuations of the membrane potential; (2) ion-activated K^+^ channels that are stimulated by the presence of Ca^2+^ or other intracellular signaling molecules; (3) inwardly rectifying K^+^ channels that transfer K^+^ more easily into the cell than out of it; (4) tandem pore domain K^+^ channels which may be constitutively open [78]. At the same time, voltage-gated and ion-activated K^+^ channels may share overlapping properties. Evidence gathered from previous studies supports the presence of several K^+^ channels in spermatogenic cells as well as in spermatozoa [79,80,81,82,83,84,85].

The calcium-activated potassium channel (SLO1) and the potassium channel subfamily U member 1 (SLO3) are the most frequently described K^+^ channels in male gametes and are considered as primary regulating channels of K^+^ currents. They share characteristics of voltage-gated and ion-activated K^+^ channels since they may be triggered by membrane depolarization, Ca^2+^ and Mg^2+^ [86,87]. Besides the K^+^ current regulation, SLO channels are involved in the regulation of osmolality and membrane potential of the plasma membrane [88]. The structure of both consists of four pore-forming α subunits and several auxiliary subunits [86,89]. SLO1 channels are present in all multicellular, mitochondrial eukaryotes, primarily in muscle and neural cells, while SLO3 is exclusive to mammalian testes and spermatozoa [83,86,90], although its isoforms may be found in the kidneys, brain, and eyes [91]. Although both channels are primarily defined as voltage gated, SLO1 is also activated by Ca^2+^ [92], while SLO3 is triggered by intracellular alkalinization [86,93].

SLO3 is the principal K^+^ channel in mammalian spermatozoa [88,94] and is localized in the principal piece of the sperm flagellum [95]. SLO3 channels are responsible for K^+^ efflux, and subsequent membrane hyperpolarization, which then affect other voltage-sensitive ion channels such as CatSper and voltage-gated calcium channels (VGCCs) [96]. Additionally, it has been speculated that SLO3 may participate in volume control and the ability of spermatozoa to respond to osmotic challenges during their transit within the female reproductive tract [97]. Because of rapid evolution, the channel presents with high structural divergence and various functional properties amongst mammals, resulting in different voltage ranges for the activation, sensitivity to pH, Ca^2+^ or phosphatidylinositol 4,5-bisphosphate and subsequent dynamics of SLO3 in different mammalian species [95,98,99].

Male SLO3 knockout mice produce spermatozoa with a reduced motility and abnormal morphology, most likely due to a lack of membrane hyperpolarization, activation of other voltage-gated channels and subsequent disturbance of osmotic homeostasis, since apparently no other channels can compensate for the loss of SLO3 [94]. Although an SLO3 mutant sperm can, to some degree, undergo spontaneous AR, it fails to undergo this exocytotic event when exposed to solubilized zona pellucida (ZP). This result supports the hypothesis that membrane hyperpolarization during capacitation is a key factor required for the induction of the AR [100]. As such, these animals are neither capable of producing offspring even during extended mating periods nor achieve fertilization in vitro [101].

Since K+ concentrations vary significantly in aquatic environments [102], the primary channels responsible for K^+^ transport and subsequent sperm plasma membrane hyperpolarization in aquatic animals are the cyclic nucleotide-gated K^+^ channels (CNGKs). Depending on the species, these tetrameric channels may be located at the head [103] or flagellum [104] and are activated by oocyte-derived chemoattractants [7]. The presence of a chemoattractant leads to an increase in cGMP [105], which opens CNGK channels, to trigger membrane hyperpolarization, followed by a continued depolarization [104,105]. Furthermore, CNGK channels play a pivotal role in the induction of sperm motility, as their activation leads to Ca^2+^ influx carried out by voltage-sensitive Ca^2+^ channels [103,104,105].

Inward rectifier K^+^ channels conduct larger inward currents at membrane voltages negative to the K^+^ equilibrium potential than outward currents at positive voltages, which enables them to be active at negative voltages [106,107]. Two types of inward rectifiers have been identified in male reproductive cells, specifically K^+^ channels with strong inward rectification and weakly rectifying K^+^ channels. K^+^ channels with strong inward rectification are highly selective to K^+^ and may be inhibited by intracellular acidification and Ba^2+^, which is a known inhibitor of sperm capacitation and acrosome reaction [108]. Weakly rectifying K^+^ channels sensitive to ATP (KATP channels) comprise Kir 6.1, Kir 6.2 (subunits of the KATP channel), SUR1 and SUR2 (sulfonylurea receptor) channels which were detected in both spermatogenic cells and mature spermatozoa, specifically in the flagellum principal piece (SUR2) and midpiece (Kir 6.1, Kir 6.2, SUR1, SUR2) as well as in the postacrosomal region of the sperm head (Kir 6.2, SUR1) [79,109]. These channels are particularly sensitive to the KATP channel blockers, tolbutamide and glibeclamide, and the loss of glucose leading to the reduction in ATP [79].

Delayed outward voltage-dependent K^+^ channels are a family of K^+^ channels that enable a sustained K^+^ efflux with a delay following membrane depolarization, which leads to a rapid membrane repolarization. There are two types of delayed rectifiers in spermatogenic cells depending on their sensitivity to tetraethyl ammonium (TEA). The most prominent TEA-sensitive channel found in spermatogenic cells and mature spermatozoa is K_V_3.1, whilst the predominant delayed rectifier less sensitive to TEA is hypothesized to correspond to the SLO3 K^+^ channels which have been discussed previously. Other less prominent delayed rectifiers identified in male reproductive cells are K_V_1.1, K_V_1.2, and G protein-coupled inwardly rectifying potassium (GIRK1) channels [80].

**Table 5 biomedicines-11-02519-t005:** Potassium channels.

Channel	Species	Localization	Functionality	References
SLO1/3	mammals including human, reptiles, birds, fish	principal piece of sperm flagellum	regulation of K^+^ efflux, osmolality and membrane potential—hyperpolarization	[86,90,94]
GNGK’s	sea urchin, zebrafish	sperm head and flagellum	induction of sperm motility, mediators of voltage sensitive Ca^2+^ channels	[103,106]
KATP	mice, rat	postacrosomal region of sperm head, midpiece and principal piece of sperm tail	capacitation-associated hyperpolarization	[78,79]
K_V_	mice (Kv1.1, Kv1.2 and Kv1.3), bull (Kv1.1)	equatorial segment of spermatozoa	membrane hyperpolarization, capacitation and acrosome reaction, maintaining of sperm osmotic resistance	[34,110]

## 7. Chloride Channels

In comparison to previously discussed ion channels and transporters, information on chloride (Cl^−^) channels is relatively sparse (Table 6). Nevertheless, it has been previously reported that sperm capacitation, hyperactivation and fertilization is blocked in media lacking Cl^−^ channels, which supports the hypothesis that regulation of Cl^−^ homeostasis is necessary for a proper membrane hyperpolarization, tyrosine phosphorylation and cytosolic alkalization [7,111,112]. According to Santi [78], Cl^−^ transporters may be divided into two categories: (1) Cl^−^ channels located in the plasma membrane with four structural families, specifically γ-aminobutyric (GABA)-gated and related glycine-gated neurotransmitter receptors, cystic fibrosis transmembrane conductance regulator (CFTR), Ca^2+^-activated Cl^−^ channels (CaCCs) and ClC-3 channels; and (2) Cl^−^ transporters which enable Cl^−^ passage with another ion in either the same or the opposite direction (symporters, cotransporters or antiporters). The most prominent Cl^−^ transporters include the electroneutral cation-Cl cotransporters such as the Na^+^/Cl^−^ cotransporter, the Na^+^/K^+^/2Cl^−^ cotransporter or the Na^+^-independent K^+^/Cl^−^ cotransporter.

Gamma-aminobutyric acid (GABA) receptors are Cl^−^ channels that are most known to mediate inhibitory neurotransmission in the central nervous system. In spermatozoa, GABA receptors have been shown to be involved in the induction of acrosome reaction [113], regulation of sperm motility and hyperactivation [114,115], as well as modulation of the response of male gametes to progesterone [114,115,116,117].

The cystic fibrosis transmembrane conductance regulator (CFTR) is a unique member of the ATP-binding cassette (ABC) transporter family that acts as an ion channel modulated by cAMP/PKA and ATP [118]. The channel is in charge of Cl^−^ and HCO_3_^−^ transport in an electrochemical gradient contrary to other members of the ABC family that transport substrates against their chemical gradients [119]. Structurally, CFTR contains two membrane-spanning domains (MSDs), two nucleotide-binding domains (NBDs), and one regulatory (R) domain. MSDs constitute the channel pore; phosphorylation of the R domain determines the activity of CFTR and ATP hydrolysis by NBDs affects the channel-gating properties [120]. CFTRs have been localized in the midpiece of mammalian spermatozoa [23,111,121] where they are involved in the regulation of sperm motility, cAMP production and membrane hyperpolarization [121]; their specific roles depend on the species. In humans, CFTRs seem to be involved in Cl^−^ removal from spermatozoa upon capacitation [33], whilst in mice and guinea pigs, CFTRs are suggested to transport Cl^−^ to spermatozoa [21,23]. In addition to its role as an Cl^−^ channel, CFTR is also known to interact with, cooperate with and regulate other ion channels such as chloride anion exchanger SLC26A3, a HCO_3_^−^ transporter of the SLC26 family, or epithelial Na^+^ channels (ENaC) [122,123,124].

Ca^2+^-activated Cl^−^ channels (CaCCs) are stimulated by increases in intracellular Ca^2+^ levels caused either by its influx through the plasma membrane or release from intracellular stores. These are anionic channels belonging to the anoctamin family (ANO/TMEM16). Depending on the species, CaCCs may be found in the sperm head (humans), the apical part of the acrosome or the middle piece of the flagella (guinea pigs) [125,126]. Evidence gathered from currently available reports suggests that CaCCs play an important role in the process of capacitation and acrosome reaction, as well as in the regulation of sperm motion, particularly in the acquisition of hyperactivated motility [7,127].

Chloride channels (ClCs) are an evolutionary conserved voltage-gated channel family of nine members found in prokaryotic as well as eukaryotic organisms [128,129]. Out of these, ClC-3, an intracellular voltage-dependent electrogenic 2Cl^−^/H^+^-exchanger [130,131], has been detected in the sperm flagellum (humans, rhesus monkeys) [90,94], as well as in the acrosome and midpiece (bulls) [132,133]. Chloride channel-3 (ClC-3) regulates outwardly rectifying Cl^−^ currents that are inhibited by protein kinase C (PKC) activation [134], chloride channels thus playing important roles in the regulation of sperm volume and motility [135]. At the same time, ClC-3 can bind protein phosphatase PP1γ2, which is crucial for sperm maturation and motility [132]. Accordingly, spermatozoa from asthenozoospermic patients present with a lower cell volume and mobility, which correlates with lower expression levels of ClC-3 [135].

**Table 6 biomedicines-11-02519-t006:** Chloride channels.

Channel	Species	Localization	Functionality	References
CFTR	mice, guinea pig, human	equatorial segment of spermatozoa	transportation of Cl^−^ to (rodents) and out (humans) of spermatozoa, intracellular alkalinization, cAMP synthesis, membrane hyperpolarization, cooperation with SLC26 channels	[21,23,112]
CaCC	human, guinea pig	sperm head, apical part of the acrosome, middle piece of sperm tail	regulation of sperm movement and acquisition of hyperactivation	[125,126]
ClC	human, monkey, bull	sperm flagella, acrosome	regulation of cell volume, capacity and mobility	[131,132,135]

Previous research has unraveled that CFTR inhibitors affect the plasma membrane hyperpolarization without compromising other aspects of capacitation (such as tyrosine phosphorylation), suggesting the presence of other Cl^−^ transporters in spermatozoa. Cl^−^ may enter the cell with the help of electroneutral carriers; specifically, the sodium–chloride symporter and the sodium–potassium–chloride carriers transport Cl^−^ into the cell, while potassium–chloride cotransporters (KCCs) transport Cl^−^ out of the cell under physiological conditions [136]. Cl^−^ levels have been shown to be increased during capacitation [111,137], indicating that NCC and NKCC may be involved in the regulation of Cl^−^ homeostasis during sperm activation and preparation for the physiologically induced acrosome reaction. Further research has revealed the presence of NKCC1 in spermatids, and null mutants of this protein present with a defective spermatogenesis and infertility [138].

Cl^−^ may be carried through the plasma membrane with the help of molecules that exchange Cl^−^ for HCO_3_^−^ in either direction. The role of HCO_3_^−^ in the activation of cAMP synthesis via soluble adenylyl cyclase is undeniable [139,140]; specific HCO_3_^−^ carriers have not yet been fully defined. Whilst previous research has identified that Na^+^/HCO_3_^−^ cotransporters are responsible for initial HCO_3_^−^ influx into the sperm cell [141], Cl^−^/HCO_3_^−^ exchangers have been suggested to be involved in the regulation of HCO_3_^−^ homeostasis. In addition, through their contribution to the Cl^−^ gradient, they are important players in the regulation of cell volume, intracellular pH, and membrane potential. The most relevant Cl^−^/HCO_3_^−^ exchangers include two evolutionary independent gene superfamilies, *SLC4* and *SLC26*, with specific patterns of anion selectivity and tissue distribution. From the SLC4 superfamily, only AE2 is found in testicular germ cells with suggested roles either in the spermatogenic process or later in sperm function. With respect to the SLC26 superfamily, recent studies have identified SLC26A3 and SLC26A6 in the sperm midpiece [21,23,78]. Accordingly, a SLC26A3-specific inhibitor blocked the capacitation-associated membrane hyperpolarization and the ZP-induced acrosome reaction, without affecting the cAMP pathway or tyrosine phosphorylation [78].

## 8. Aquaporins

Aquaporins (AQPs) are a ubiquitous transmembrane protein family (Table 7) playing pivotal roles in cellular fluid homeostasis, facilitating bidirectional water diffusion across the membrane [142,143]. Structurally, AQPs are formed by four monomers, each with their own permeable pore and with one central pore inside the tetramer whose function is currently unknown [144]. Thirteen AQPs are currently known with different permeability properties, structural features, and localization [145]:Classical or orthodox AQPs present with the smallest channel size and hydrophilic nature. Orthodox AQPs are located in the plasma membrane and are considered primarily selective to water (with the exception of AQP6 present in the membranes of cellular organelles that acts as a selective anion channel). This group comprises AQP0, AQP1, AQP2, AQP4, AQP5, AQP6, and AQP8, considered primarily selective to water [144,146,147].Aquaglyceroporins (GLPs) present with larger pore size and lower hydrophilicity. As such, these proteins are able to permeate glycerol (preferably), arsenite, urea, polyols, purines, or pyrimidines. AQP3, AQP7, AQP9, and AQP10 belong to this group [142,148,149].Nonorthodox AQPs or superaquaporins (superAQPs) include AQP11 and AQP12. These are expressed in intracellular membranes, primarily in the endoplasmatic reticulum. While it has been reported that superAQPs are involved in the transport of water and glycerol, their specific pore selectivity and function are currently unknown [142,150].

All AQPs, except for AQP6 and AQP12, have been found in different locations of the male reproductive system as well as in spermatozoa, out of which AQP3, AQP7, AQP8 and AQP11 are the most prominent ones [144,151]. Their specific location by and large differs among species. AQP3 is present in the sperm mid-piece in bulls and boars [152,153], while in human and murine sperm, the principal piece is its prime location [154,155]. AQP7 is present in the tail of ejaculated spermatozoa in bulls, stallions, boars, mice, and rats, as well as in certain regions of the sperm head of men [153,156,157,158,159,160]. AQP8 has been detected in the tail of human, mouse, and rat spermatozoa, additionally to the mitochondria from the mid-piece in men [159,161]. Finally, AQP11 is present in the intracellular tail structures of human, boar, stallion, mouse, and rat spermatozoa. In the meantime, its presence was also confirmed in the sperm head of humans and the terminal piece of the sperm tail in rats [157,162,163,164].

The most important role of AQPs in male gametes is related to osmoregulation to counteract hypoosmotic stress that spermatozoa must withstand upon entering the female reproductive tract [142]. The resulting hypoosmotic shock causes spermatozoa to uptake excessive amounts of water, leading to swelling, membrane ruptures and the loss of proper movement of the sperm tail [143]. Under physiological conditions, hypoosmotic stress triggers an osmolyte efflux that drives a rapid water trafficking via AQPs and restores cell volume [76]. Nevertheless, osmotic changes that occur in response to the hypotonic shock that spermatozoa experience are crucial to initiate the capacitation process. Changes in the sperm volume trigger the opening of calcium channels which enable calcium influx, which is the first event occurring during capacitation [144]. At the same time, acrosomal swelling is an essential prerequisite for a physiological acrosome reaction [145]. In the meantime, evidence from recent years points to the necessity of strictly regulated hydrogen peroxide (H_2_O_2_) levels, which are essential for sperm capacitation, hyperactivation and acrosome reaction [146]. Since some AQPs (particularly AQP8) have been suggested to be involved in the diffusion of H_2_O_2_, their possible involvement in ROS-mediated sperm activation seems plausible. Accordingly, experimental blockade of AQPs has been reported to impede ROS detoxification, leading to excessive intracellular ROS accumulation with an inhibitory effect on the sperm capacitation accompanied by plasma membrane ruptures, low membrane hyperpolarization and premature acrosome exocytosis. Finally, the permeability of AQPs to glycerol has been revealed to be important for the use of this molecule in metabolic pathways and as a source of energy [147].

**Table 7 biomedicines-11-02519-t007:** Aquaporins.

Channel	Species	Localization	Functionality	References
AQP3	bulls, boars, human, murine	midpiece and principal piece of sperm tail	osmotic balance as a defense to hypoosmotic stress, restoring of cell volume for Ca^2+^ influx and triggering of capacitation process, diffusion of H_2_O_2_	[153,156,157,158,160,161,163,164,165,166,167,168,169,170]
AQP7	bulls, stallions, boars, mice, rats, human	sperm flagella and head (human)		
AQP8	boars, stallions, mice, rats, human	spermatozoa tail		
AQP11	boars, stallions, rodents, human	intracellular tail structure, terminal piece (rats) and sperm head (human)		

## 9. Ion Channels Relevant to the Sperm Redundant Nuclear Envelope (RNE)

The redundant nuclear envelope (RNE) is defined as a residual nuclear membrane that accumulates at the sperm neck due to nuclear condensation. This membrane is considered as a continuum of the membrane covering the endoplasmic reticulum of the sperm cell before entering spermiogenesis, and occasionally RNE carries the remains of a functional endoplasmic membrane [171]. From a functional point of view, RNE has been proposed to contribute to the generation of Ca^2+^ signals necessary for sperm activation [172]. Accordingly, Ho and Suarez [171] unraveled the presence of a receptor-operated Ca^2+^ channel (IP3R) as well as a Ca^2+^-binding and storage protein called calreticulin in the region occupied by RNE. Subsequent functional experiments have revealed that IP3R mobilizes Ca^2+^ in the sperm neck, leading to an efflux of Ca^2+^ from RNE to trigger hyperactivated motility [173,174]. Moreover, Naaby-Hansen et al. [175] localized IP3R and calreticulin in the equatorial segment of the acrosome, in vesicles in the sperm neck close to the nucleus and in the cytoplasmic droplet.

## 10. Ion Channels Relevant to the Sperm Mitochondria

The integrity of mitochondrial membranes is a crucial prerequisite for proper mitochondrial function. Accordingly, mitochondrial ion channels present in the outer as well as inner mitochondrial membrane are recognized as essential regulators of mitochondrial function [176,177] and may be divided into two major types. Channels of the first type, including most K+ channels, present with properties similar to those located in the plasma membrane. Channels of the second type are exclusive for mitochondria, such as mitochondrial calcium channels [155] or mitochondrial porins [178,179].

Mitochondrial porins are voltage-dependent anion channels (VDACs) located in the outer mitochondrial membrane whose primary function is to regulate the exchange of ATP/ADP, Ca^2+^, and other metabolites and/or ions between the cytoplasm and mitochondria [180]. Two isomers, specifically VDAC2 and VDAC3, have been identified in the head, acrosome, and outer dense fibers of the flagellum in bovine spermatozoa [181] where they control the cross-talk between mitochondria and the rest of the cell. Their crucial involvement in the regulation of the sperm fertilization ability is furthermore evidenced by Kwon et al. who observed that locking VDAC significantly decreases motility, viability, acrosome reaction, capacitation, tyrosine phosphorylation, fertilization, and embryo development, regardless of Ca^2+^ levels [182].

Mitochondria enable a rapid uptake of Ca^2+^ into the matrix and thus are involved in the regulation of cytosolic Ca^2+^ signals [183]. The molecular machinery underlying Ca^2+^ uptake into energized mitochondria is driven through the mitochondrial calcium uniporter (mCU) complex (mCUC). It is a highly selective ion channel, mediating the Ca^2+^ influx across the inner mitochondrial membrane driven by negative mitochondrial membrane potential (ΔΨm) [184]. Experimental studies have revealed that a proper mCUC function is a prerequisite for a desirable sperm viability, motility, and ATP levels while sustaining a proper ΔΨm and ROS production [185].

Potassium is a crucial element for the mitochondrial integrity as its cycle regulates the mitochondrial volume and homeostasis [186]. The K^+^ influx into mitochondria is driven by a negative ΔΨm, accompanied by anion and water flux, which leads to mitochondrial swelling. The first K+ selective mitochondrial channel to be described is mitoKATP, driven by ATP and localized in the inner mitochondrial membrane [187]. The channel shows similarities with channels regulated by ATP present in the plasma membrane [188]. In the meantime, the most known mitochondrial Ca^2+^-activated K^+^ channels include the small conductance K^+^ (mitoSKCa) channel [189], the intermediate-conductance Ca^2+^-activated K^+^ channel (mitoIKCa) [190] and the large-conductance Ca^2+^-activated K^+^ channel (mitoBKCa) [191]. The latter has received increased attention for its wide occurrence in various cell types and suggested participation in cytoprotection [192]. Activation of the channel leads to an influx of K^+^ into the mitochondrial matrix, followed by membrane depolarization, and a decrease in ROS production [193]. Among voltage-gated potassium channels, the inner mitochondrial membrane holds (a) the mitochondrial 1.3 voltage-gated potassium (mitoKv1.3) channel [192]; (b) the mitochondrial 1.5 voltage-gated potassium (mitoKv1.5) channel; and (c) the mitochondrial 7.4 voltage-gated potassium (mitoKv7.4) channel [194]. In isolated mitochondria, modulation of both mitoKv1.3 and mitoKv7.4 leads to changes in ΔΨm and ROS levels, suggesting that both channels are open under physiological conditions [192]. The closure of mitoKv1.3 by either inhibitors or apoptotic regulators leads to excessive ROS release, most likely due to interactions between mitoKv1.3 and the respiratory complex I [195].

Besides cation-selective channels, the inner mitochondrial membrane also contains anion-selective transport systems [196], such as the inner membrane anion channel IMAC, also called the mitochondrial Centum picoSiemens (mCS), which is involved in the regulation of the mitochondrial volume homeostasis [197]. More information is available on channels belonging to chloride intracellular channel proteins (CLIC), specifically CLIC4 and CLIC5, which were recently detected in mitochondrial membranes [198] and are likely to play significant roles in sperm function [199].

Since mitochondria are able to uptake large amounts of Ca^2+^, a fast pathway for Ca^2+^ release is associated with mitochondrial permeability transition pore (MPTP), which contributes to cellular homeostasis and prevents mitochondria from Ca^2+^ overload [200]. MPTP is a multiprotein complex whose opening triggers a massive increase in the inner mitochondrial membrane permeability to solutes of up to 1.5 kDa [201]. According to existing evidence, several proteins present in both mitochondrial membranes and the matrix have been associated with MPTP, including the VDA channels, BCL-2 proteins, adenine nucleotide translocator (ANT) or hexokinase [202,203,204]. In the meantime, the opening of the pore is accelerated by the loss of the inner ΔΨm [205]; the alkalinization of the mitochondrial matrix [206]; and the increase in Ca^2+^ levels [207]. A transient MPTP opening in intact and healthy cells contributes to cellular homeostasis since it provides a fast release of ions or toxic compounds accumulated in the mitochondrial matrix [208]. Moreover, an MPTP opening may also regulate the activity of some mitochondrial Ca^2+^-dependent enzymes [209].

## 11. Ion Channels Relevant to the Sperm Acrosome

The acrosome is a membrane-derived organelle that covers the sperm head of numerous species. Functional acrosomal structures are a critical component of the sperm–egg fusion that is triggered by physiological inducers released from the female gamete or by exposure to specific pharmacological stimuli. Acrosome reaction is defined by a strictly regulated and irreversible process during which the acrosomal contents including Ca^2+^ and enzymes are released to the extracellular medium [210]. The activation of the acrosome reaction relies on the concurrence of several transduction pathways, most notably ion permeability changes leading to increased intracellular pH, Ca^2+^ and cAMP, G protein activation, changes to the membrane lipids and protein phosphorylation. Since several ion channel blockers inhibit the progress of acrosome reaction, the role of membrane transporters in these secretory events is indisputable [43,211]. Hence, this section briefly discusses the most important ion channels that are relevant for a proper acrosome functionality.

Voltage-gated Ca^2+^ channels (VGCCs) have been found in the sperm acrosome of a broad range of species including fish [58], newts [57], and mammals [60,212]. Several types of VGCCs have been identified in spermatozoa, specifically the “long-lasting” L-type that is activated by high voltage and is resistant to ω-conotoxin and ω-agatoxin [58,60,212]; the “transient” T-type channel operated by low voltage [212,213]; and the “Purkinje” P/Q-type that is activated by high voltage, is resistant to ω-conotoxin and blocked by ω-agatoxin [212]. Both the L- and T-type channels have been reported to participate in human sperm acrosome reactions [60] by activating membrane depolarization and mediating Ca^2+^ influx in response to changes in the action potential and depolarizing signals [214].

Transient receptor potential vanilloids (TRPVs) consist of six subtypes divided into two groups depending on their Ca^2+^ permeability and sensitivity to temperature, specifically TRPV1/TRPV2/TRPV3/TRPV4 and TRPV5/TRPV6 [215]. All six TRPV subtypes have been found in the spermatozoa of vertebrates, even though TRPV1 and TRPV4 seem to be present more frequently. TRPV1 is a voltage/heat/lipid/pH-modulated channel localized in the sperm head and the acrosome [45,216,217] which is desensitized by internal Ca^2+^; however, the channel is not activated by Ca^2+^-store reduction [215]. The intensity of the TRPV1 current rises with increasingly acidic pH and is regulated by intracellular phosphatidylinositol 4,5-bisphosphate [215]. In the meantime, the TRPV4 channel activation depends on the extracellular osmolality, pH, lipids, and mechanical triggers, such as shear stress or membrane stretching [215,218].

Store-operated Ca^2+^ channels (SOCCs) are inward rectifiers, and their primary role is to supply the cellular compartments with Ca^2+^ from extracellular environment after Ca^2+^ is released and pumped out across the plasma membrane [43]. Since SOCCs are not voltage-dependent channels, they are functional even at negative membrane potentials at which depolarization-sensitive channels (such as VGCCs) are not engaged in action. SOCCs are assembled by ORAI1–3 proteins, where one channel is created by one ORAI protein. The channel is inactivated by Ca^2+^ binding [48]. SOCs have been suggested to play an important role in the regulation of sperm physiology as the channel inhibition reduces sperm motility and acrosome reaction [51,219]. This phenomenon is most likely regulated through the induction of 5′ AMP-activated protein kinase phosphorylation [51].

Similar to the sperm plasma membrane, CatSper channels are the most studied sperm Ca^2+^ channels in the acrosome structures because of their sperm specificity and crucial roles in sperm–egg chemotaxis, capacitation, and acrosome reaction [38]. Besides Ca^2+^, CatSper also facilitates the entry of monovalent (Na^+^ and Cs^+^) and bivalent cations (Ba^2+^) to spermatozoa if extracellular Ca^2+^ is absent. The channel is pH-sensitive and triggered by alkaline pH [38]. Its activity is furthermore regulated by cyclic nucleotides, membrane voltage, phosphorylation, biomolecules (such as prostaglandin, BSA and progesterone) and zona pellucida glycoproteins [38,220].

A Ca^2+^-activated Cl^−^ channel (CaCC) opening is stimulated by increases in intracellular Ca^2+^ levels resulting from its influx through the plasma membrane channels or its release from intracellular stores. CaCCs have been reported to be present in the heads of mature human spermatozoa where they may contribute to Ca^2+^-dependent Cl^−^ currents necessary for a proper acrosome reaction [130]. In the meantime, chloride channels (ClCs), specifically ClC-3, have been detected in the acrosome and midpiece of bovine spermatozoa [133], playing important roles in the regulation of sperm volume, motility, and fertilization ability [135].

The main role of a voltage-gated H^+^ channel (VGHC) is to extrude H^+^ from the cell, leading to an increase in intracellular pH [221]. These channels exhibit a highly selective H^+^-conductance, opening with membrane depolarization, high extracellular pH and decreased intracellular pH [221]. Besides in humans [76,222,223], VGHCs have also been found in macaque [224], boar [225], and bull spermatozoa [61]. VGHC activation leads to intracellular alkalinization, which is accompanied by CatSper activation, Ca^2+^ influx and the induction of hypermotility and acrosome activation [213,222,226]. As such, VGHC has been shown to be involved in the induction of capacitation, progressive motility, and acrosome reaction through induced NADPH oxidase 5 activation and ROS generation [69,222,227]. Correspondingly, their inhibition leads to a reduced sperm motility and progesterone-induced acrosome reaction [225,226].

## 12. Conclusions

Regulation machinery behind the whole capacitation process is still not fully understood. However, what we can say for sure is that ion channels of the sperm plasma membrane are responsible for the maintenance of various biological and biochemical changes such as alkalinization, hyperpolarization, hyperactivation as well as capacitation. These channels and transporters supported the adaptation of sperm cells to a constantly changing environment during their fertilization journey in the female genital tract. Accordingly, since their dysfunction has been frequently correlated with sub- or infertility, a more profound understanding of their involvement in the regulation of sperm behavior in future studies may contribute to the evolution of new strategies for the management of male reproductive dysfunction.

## Figures and Tables

**Figure 1 biomedicines-11-02519-f001:**
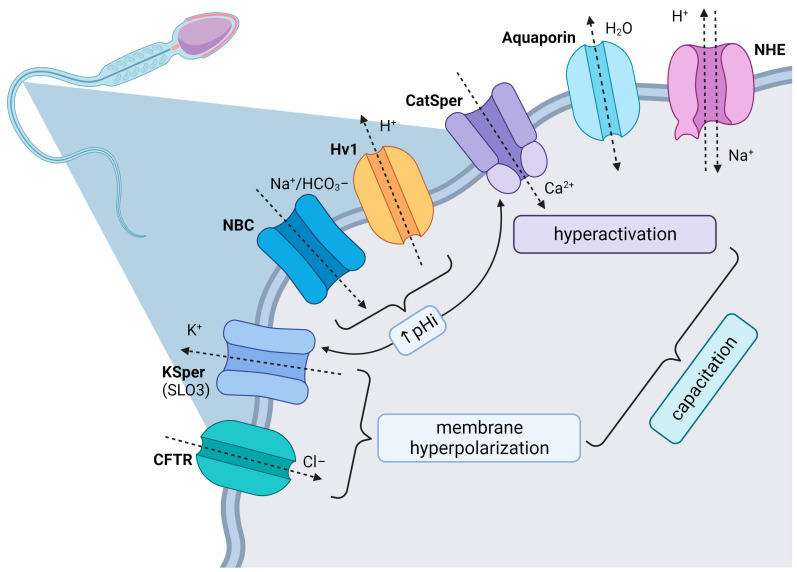
Ion channels of spermatozoa. Created with BioRender.com (Toronto, ON, Canada, https://app.biorender.com/).

## Data Availability

No new data were created or analyzed in this study. Data sharing is not applicable to this article.

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
