# Peer review of "Signaling Roleplay between Ion Channels during Mammalian Sperm Capacitation"

_biomedicines, 2023, doi:10.3390/biomedicines11092519_

Round 1

Reviewer 1 Report

1. In order to get a complete picture of sperm ion channels, more information should be provided regarding ion channels in intracellular membranes like the acrosome, mitochondria and redundant nuclear envelope(RNE).

2. Line 139: intracellular Ca concentrations is also regulated by Ca-pumps of the plasma and outer acrosomal membranes and mitochondrial transporters.

3.Line 188-197: SOCC is activated when intraacrosoml [Ca] is reduced due to Ca efflux from the acrosome via IP3-R. 

Author Response

Respond to Reviewer 1 Comments 

Thank you very much for taking the time to review this manuscript. Please find the detailed responses below and the corrections highlighted yellow in the re-submitted files.

Comment 1: In order to get a complete picture of sperm ion channels, more information should be provided regarding ion channels in intracellular membranes like the acrosome, mitochondria and redundant nuclear envelope(RNE).

Answer 1: Thank you for your idea. We added a new chapters about the topics you seggested in the manuscript. You can read them in the attached file. 

Comment 2: Line 139: intracellular Ca concentrations is also regulated by Ca-pumps of the plasma and outer acrosomal membranes and mitochondrial transporters.

Answer 2: Thank you for the correction. We found the sentence and change it. 

Comment 3: Line 188-197: SOCC is activated when intraacrosoml [Ca] is reduced due to Ca efflux from the acrosome via IP3-R. 

Answer 3: We fixed the mistake. Thanks.

Please see the whole revised manuscript in the attachment. 

Best regards, 

Dr. Benko and co-authors of the manuscript. 

Reviewer 2 Report

In this Manuscript the authors review the presence and function of ion channels in the sperm plasma membrane. This is not a new topic, but the review is well written. There are a few issues the authors must consider in a revision in terms of manuscript organization:

-In the manuscript (title, abstract, etc.) the authors mention only “sperm”, which is not accurate. It is clear that the authors mean “mammalian sperm”, and this change should be made at the beginning.

-The authors should describe how they performed their literature search for this systematic narrative review. What terms were used, what was the time frame, which papers were included/excluded (this should be supplemental).

-The initial figure is useful, but them there is just narrative text, which makes reading the paper extremely tiring and not very motivating for a reader. The authors should create Tables for each of the channels described, noting the most important papers that revealed their presence/function, what species the experiments were performed, the main conclusions of the studies in telegraphic form etc.

Author Response

Respond to Reviewer 2 Comments 

Thank you very much for taking the time to review this manuscript. Please find the detailed responses below and the corrections highlighted yellow in the re-submitted files.

Comment 1: In the manuscript (title, abstract, etc.) the authors mention only “sperm”, which is not accurate. It is clear that the authors mean “mammalian sperm”, and this change should be made at the beginning.

Answer 1: Thank you for your perseption. We changed the title of the review as follow: Signaling Roleplay between Ion Channels during Mammalian Sperm Capacitation.

Comment 2: The authors should describe how they performed their literature search for this systematic narrative review. What terms were used, what was the time frame, which papers were included/excluded (this should be supplemental).

Answer 2: Thanks a lot for the recommendation. We write a sentence about review methodology and search strategy. There is a sentence about review methodology and search strategy, which can be later upload as a supplementary material as you suggested: 

Review Methodology 

Search Strategy 

A systematic review of the literature was carried out on June 2, 2023 by submitting selected keywords into three different databases: PubMed, Scopus, and Web of Science. Search terms used included “Ion channels”, “Bicarbonate channels” or “HCO3- channels”, “Sodium channels” or “Na+ channels”, “Calcium channels” or “Ca2+ channels”, “Proton channels” or “H+ channels”, “Potassium channels” or “K+ channels”, “Chloride channels” or “Cl- channels”, “Aquaporins”, “Spermatozoa” or “Sperm”, “Capacitation”. A time filter was applied to the search to isolate solely the works published in the last 20 years (2003–2022). 

Search Eligibility Criteria 

The papers collected from the above searches were screened for the presence of duplicates and narrowed down further using the predefined inclusion and exclusion criteria based on the journal title and abstract. Selected exclusion criteria were (1) the paper is not written in English, (2) the paper is not an original research paper (review articles, book chapters, editorials, conference abstracts, and letters; studies with no abstracts were eliminated), or (3) the paper is not freely available (using the institutional credentials of the Slovak University of Agriculture). The resulting list was then subjected to an inclusion round in which we considered only original laboratory research studies, conducted on spermatozoa as a model, that met our original search aims. All papers that did not discuss the involvement of ion channels in sperm activation in the title or abstract were excluded. In summary, the inclusion criteria were (1) the paper covers original laboratory research; (2) spermatozoa are the model of the study; and (3) the paper is relevant based on its title and abstract. 

Article Selection and Processing 

The final list obtained by this iterative selection process was independently reviewed by three different co-authors of the paper. The full text was obtained for each of the included articles. If an article was not readily available, the corresponding author was personally contacted, and the manuscript of the relevant study was obtained. No papers were excluded in this step. 

Search Results 

The initial search yielded 800 titles from PubMed, 426 titles from Scopus, and 611 titles from Web of Science, for a total of 1 837. The search results were collated, and subsequent removal of duplicates reduced this number to 1 269. The exclusion round led to the rejection of 436 reviews, editorials, book chapters, articles not written in English or unavailable papers. The final round of rejection eliminated 284 articles that did not meet the inclusion criteria. The final number of articles that had met the pre-established eligibility criteria was and the subsequent exclusion and inclusion rounds led to the final number of 549 articles with information that was considered for the purposes of this review. 

Comment 3: The initial figure is useful, but them there is just narrative text, which makes reading the paper extremely tiring and not very motivating for a reader. The authors should create Tables for each of the channels described, noting the most important papers that revealed their presence/function, what species the experiments were performed, the main conclusions of the studies in telegraphic form etc.

Answer 3: Thank you for your idea. At the end of every chapter we added a table with the information about ion channels, which include their localization, functionality and other characteristics with linked references.  

Please see the whole revised manuscript in the attachment. 

Best regards, 

Dr. Benko and co-authors of the manuscript. 

Round 2

Reviewer 2 Report

The authors have adequately addressed ll my comments. I have no further issues